# Pre-Carpels from the Middle Triassic of Spain

**DOI:** 10.3390/plants11212833

**Published:** 2022-10-25

**Authors:** Artai A. Santos, Xin Wang

**Affiliations:** 1Departamento de Xeociencias Mariñas e Ordenaciόn do Territorio, Universidade de Vigo, 36200 Vigo, Spain; 2State Key Laboratory of Palaeobiology and Stratigraphy, Nanjing Institute of Geology and Palaeontology and CAS Center for Excellence in Life and Paleoenvironment, Chinese Academy of Sciences, Nanjing 210008, China

**Keywords:** origin, flowers, carpels, triassic, Spain

## Abstract

In stark contrast to the multitude of hypotheses on carpel evolution, there is little fossil evidence testing these hypotheses. The recent discovery of angiosperms from the Early Jurassic makes the search for precursors of angiosperm carpels in the Triassic more promising. Our light microscopic and SEM observations on *Combina* gen. nov., a cone-like organ from the Middle Triassic of Spain, indicate that its lateral unit includes an axillary anatropous ovule and a subtending bract, and the latter almost fully encloses the former. Such an observation not only favors one of the theoretical predictions but also makes some Mesozoic gymnosperms (especially conifers and *Combina*) comparable to some angiosperms. *Combina* gen. nov. appears to be an important chimeric fossil plant that may complete the evidence chain of the origin of carpels in geological history, partially narrowing the gap between angiosperms and gymnosperms.

## 1. Introduction

The origin of angiosperms and their relationship with other seed plants have been the foci of botanical debates for a long time [1,2]. Carpels (the basic units of gynoecia in angiosperms) are idiosyncratic to angiosperms [3]. According to the traditional theory, a carpel results from the longitudinal folding and enrolling of a megasporophyll that bears ovules along its margins [4,5,6,7]. This hypothesis sounded rational, especially when Goethe’s dictum “Alles ist Blatt” was taken into consideration [4]. However, it has been discarded since the APG system came into existence [8], and the APG system cannot give a plausible morphological interpretation for carpel homology [9]. Thus, plant systematics has entered a dead end: no widely accepted interpretation for the origin and homology of carpels is given, leaving many botanical questions unanswered. Therefore, using a fossil reproductive organ morphologically intermediate between angiosperms and gymnosperms to sift one hypothesis out of many is of crucial importance to move plant systematics beyond this debate. Although challenged by three groups of authors [10,11,12], *Nanjinganthus*, based on over 200 specimens of flowers, remains robust as an angiosperm from the Early Jurassic, since these challengers could not reach a consensus on the definition of angiosperms among themselves [13]. The Early Jurassic age of *Nanjinganthus* [13,14,15] suggests that the Triassic is a promising period for the search for a carpel precursor. Here, we report a new cone-like reproductive organ, *Combina* gen. nov. (Figure 1 and Figure 2), from the Anisian (the lower Middle Triassic, >242 Ma) of Spain. In contrast to the seed–scale–bract–complex (SSBC) frequently seen in conifer cones, each lateral unit in *Combina* gen. nov. comprises an anatropous ovule in the axil of a bract that folds longitudinally and almost fully encloses the ovule. Such a configuration demonstrates a certain resemblance to both SSBCs in some Mesozoic “conifers” and carpels in some extant basal angiosperms (e.g., *Illicium*, *Michelia*) [16,17], suggesting a possible common ancestor shared by some angiosperms and Mesozoic “conifers”. Its unique morphology and Triassic age make *Combina* gen. nov. one of the plausible ancestor candidates for angiosperm carpels, which otherwise were thought to emerge out of nowhere.

## 2. Results 

### 2.1. Systematic Palaeontology

#### 2.1.1. *Combina* gen. nov.

**Diagnosis**: Reproductive organ, cone-like and cylindrical, with helically arranged lateral units. Each lateral unit including an axillary ovule and a subtending bract. Ovule anatropous, attached to the organ axis. Bract longitudinally folded, with a ventral longitudinal suture.

**Etymology:***Combina* from the Latin word “combinare”, meaning “unite two things together”, since the fossil combines the characteristics of angiosperms and conifers. 

**Type species:***Combina triassica* gen. et sp. nov.

#### 2.1.2. *Combina triassica* gen. et sp. nov.

(Figure 1)

**Diagnosis**: In addition to the genus diagnosis: organ with at least 13 lateral units helically arranged. Lateral units decreasing distally in size. Bracts elliptical to ovate in shape, up to 10 mm long. Ovules attached to the organ axis, up to 8 mm long, with a smooth outline and an asymmetrical base.

**Description**: The organ is cylindrical in form, 38 mm long and 16 mm wide, including more than 13 lateral units (Figure 1a). The lateral units are helically arranged along the organ axis, decreasing distally in size (Figure 1a). The lateral units diverge from the organ axis at angles between 27° and 66° (Figure 1a). Each lateral unit comprises a subtending bract and an axillary ovule (Figure 1b,c,g–j). The bract is 7 to 10 mm long, elliptical to ovate in shape, almost fully enclosing the ovule from bottom and laterals, leaving a ventral gaping suture (Figure 1a–j). Each ovule is attached to the organ axis by a short funiculus (Figure 1a–f,i,j). The ovule is anatropous, 7–8 mm long, and 1–2.3 mm thick, with a funiculus about 1 mm long and 0.3–1 mm in diameter (Figure 1b,c,g–j). The ovule is smooth-outlined and basally asymmetrical (Figure 1b,c,g–j; Figure 3).

Associated with the holotype of *Combina* is a part (Figure 2a,b) that resembles the lateral units *in situ* and thus provides an otherwise unavailable perspective on the lateral unit of *Combina*. The ovule and its funiculus in the holotype of *Combina* are not fully smooth and asymmetrical (Figure 1i,j): the funiculi appear to skew to one side of the ovule, suggestive of an anatropous ovule. This observation and inference of the holotype are further confirmed by an adaxial view of the isolated lateral unit (Figure 2a,b). 

**Holotype:** MPZ2009-425. 

**Further specimen:** R4/40/38/5.

**Etymology:***triassica* for the Triassic, the age of the fossil.

**Locality:** Rodanas outcrop, Aragonian Branch of Iberian Range, Spain.

**Horizon and Age**: The Calcena Formation, Anisian, lower Middle Triassic [18].

## 3. Discussion

The general morphology of *Combina* appears to be that of a coniferous cone, in which a lateral unit is composed of an axillary scale and a subtending bract. According to Schweitzer and Kirchner [19], there are only two Mesozoic genera with unilobate one-seeded lateral units, one is *Drepanolepis* and the other *Ontheodendron*. Since *Ontheodendron* has been recognized as a fossil stem [20,21], there is only one fossil genus left for comparison, *Drepanolepis*. Although both *Combina* and *Drepanolepis* appear similar to conifer cones in inverted ovules/seeds, *Drepanolepis* figured by Schweitzer and Kirchner is much more elongated and slenderer than *Combina*; most importantly, the bract encloses the axillary ovule to an unprecedented extent and forms an adaxial suture in *Combina* (Figure 1a,d,e), which is one of the implementations of the universal evolution trend of plant and organism reproduction [22]. Hitherto, the latter has not been seen in any known conifer cones (including *Drepanolepis*). This comparison justifies *Combina* as a new genus. Despite these differences between *Combina* and *Drepanolepis*, the comparison between them is meaningful for the homology of carpel and plant systematics (see Figure 4).

Currently, there are two competing hypotheses in botany on the homology and origin of angiosperm carpels, the Traditional Theory [4,6,23] and the Unifying Theory [24,25,26]. According to the former, a carpel is derived from a megasporophyll bearing ovules along its margins through longitudinal folding and enrolling [4,6,23]. According to the latter [24,25,26], a carpel is a composite organ comprising two parts of different nature, an axillary placenta (ovule-bearing branch) and a subtending ovarian wall (leaf). Although the Unifying Theory is in line with the results of function gene studies [27,28,29,30] as well as anatomical and morphological studies [16,17,31], it requires further independent observations (especially of fossil plants) for confirmation.

It is evident that the ovules in *Combina* gen. nov. are borne directly on the organ axis (Figure 1b–j), not on the margins of any foliar part (bract), thus at odds with the widely-accepted Traditional Theory, which expects ovules on the margins of a leaf. It is noteworthy that *Combina* is not the only evidence against the Traditional Theory, as (1) the carpels in the previously assumed ancestral angiosperm, *Michelia* (Magnoliaceae) and *Illicum* (Schisandraceae)*,* have been shown to be composed of a foliar part and an axillary ovuliferous branch [17] or an ovule directly borne on the floral axis [16]; (2) in addition, another Triassic fossil reproductive organ putatively related to angiosperm, *Nubilora*, has its ovules directly borne on the floral axis [25]; (3) the carpel precursor assumed by the Traditional Theory (namely, megasporophyll) has not yet been found in either fossil or extant plants despite a century-long intensive painstaking search [32,33].

Several attempts have been made to raise hypotheses other than the traditional theory to account for the origin of carpels [3,24,25,31,34,35]. Regrettably, no plausible new morphological interpretation of carpel origin was given under the framework of the APG system, which was mainly based on molecular data. Although Sauquet et al. [1] discussed the arrangement of carpels in ancestral angiosperm flowers, they did not touch on the problem of the origin of carpels; their conclusion became controversial immediately and triggered criticism and heated debates among botanists [36,37,38]. Recent studies of extant basal angiosperms (e.g., *Illicium* [16] *and Michelia* [17]) reinforce that a carpel comprises axillary ovule(s) and a subtending foliar part, a conclusion that has long been suggested by gene function studies [29,30]. Integrating the outcomes of previous independent studies, Wang [24,25] proposed that a carpel is a composite organ derived through synorganizing an ovuliferous branch and a subtending leaf, a pattern that is workable for *Amborella* [25] (the currently assumed basalmost angiosperm [8,39]) as well as *Magnolia* (the traditionally assumed basalmost angiosperm [4,6]). The Late Triassic *Nubilora* [25] favors this generalization, as its ovules are attached directly to the cone axis and enwrapped by foliar parts. However, this single piece of fossil evidence appears weak against the overwhelming thinking inertia in botany. Now, with its axillary ovule directly attached to the organ axis and almost fully enclosed by its subtending bract, *Combina* gen. nov. seems to fit into the scenario depicted in figure 6.3e of Taylor and Kirchner [3] and the cases 23–25 in figure 8.40 of Wang [25]. The difference between *Combina* and the anticipated precursor of carpel [3] is restricted to its anatropous configuration of its ovule, which may be interpreted as a highly reduced and metamorphosed version of a former ovuliferous branch. It is noteworthy that a single axillary ovule is enclosed by a subtending leaf in *Illicium* from bottom and laterals [16], especially similar to our observation of *Combina*. Thus, *Combina* appears to be a piece of evidence supporting the Unifying Theory [25].

The most intriguing fact about *Combina* gen. nov. is that the bracts of *Combina* gen. nov. have started enclosing their axillary ovules, although not fully (Figure 1b–j). Therefore, although we are not sure whether the ovules of *Combina* gen. nov. are fully enclosed before pollination (if so, then *Combina* is a *bona fide* angiosperm) for the time being, *Combina* gen. nov. is apparently knocking on the door of angiosperms. It is noteworthy that Endress [40] recently took a carpel as the result of synorganization between a foliar part and ovule(s). This point of view is in full agreement with the implications given by *Combina* gen. nov. The possibility of the bract forming an outer integument in *Combina* can be easily excluded by the following facts: (1) the bract is directly attached to the organ axis rather than to the funiculus (ovule base) in *Combina*; (2) the bract and ovules have distinct and separated contours in *Combina* (Figure 2a,b).

At the right time and with the right morphology, *Combina* gen. nov. appears to be an ideal precursor for angiosperm carpels, as it seems to have completed the evidence chain for carpel origination from a gymnospermous ancestor. The resemblance between *Combina* gen. nov. and the Palaeozoic conifers and Cordaitales [41]) (Figure 4a–l) seems to suggest that at least some angiosperms may share a common ancestor with some “conifers” and Cordaitales, if the previous axillary ovuliferous branches (as in Cordaitales [41], *Palissya* [42,43,44] , *Metridiostrobus* [45], and *Stachyotaxus* [44,46]) are reduced into a single ovule. Although we admit that relating conifers to angiosperms is at odds with most systematists (who cannot offer a plausible solution for carpel origin, however), our current proposal appears optimal, at least in terms of carpel homology, in the current academic context. Although we cannot determine, for the time being, that *Combina* is an angiosperm or an ancestral angiosperm, and the ovules in *Combina* are not fully enclosed as in angiosperms, it is noteworthy that, among all fossil plants, *Combina* demonstrates an unprecedented way of ovule-enclosing similar to that in some angiosperms (Illicium). This information is helpful for botanists trying to piece together the picture of plant evolution.

## 4. Materials and Methods

The plants associated with *Combina triassica* gen. et sp. nov. included *Anomopteris mougeotii, Endolepis* sp., *Equisetites* sp., *Neocalamites* sp., *N.* cf. *carrerei, Pelourdea vogesiaca, Voltzia heterophylla, V. walchiaeformis, V.* sp., and *Willsiostrobus rhomboidalis*, and the palynological assemblage was characterized by the occurrence of index taxa such as *Hexasaccites muelleri, Alisporites grauvogeli, Voltziaceasporites heteromorpha*, and several forms of *Triadispora* (*T. aurea, T. crassa, T. epigona, T. falcata*, and *T. staplinii*) [18]. A previous study indicated that *Combina* gen. nov. was from the Calcena Formation, Anisian, lower Middle Triassic [18].

Photographs were taken using a Nikon D-90 camera with an AF-S Micro Nikkor 60-mm macro lens (Canon Europa N.V., Bovenkerkerweg 59 1185 XB Amstelveen, Netherland). Cross-polarized illumination was used following the technique of image acquisition described by Kerp and Bomfleur [47]. Photographs from Scanning Electron Microscope (SEM) were taken with a JEOL JSM6010LA at CACTI (Centro de Apoio Científico-Tecnolóxico á Investigación, University of Vigo, Ourense, Spain). The pictures were organized for publication using Photoshop 7.0.

## Figures and Tables

**Figure 1 plants-11-02833-f001:**
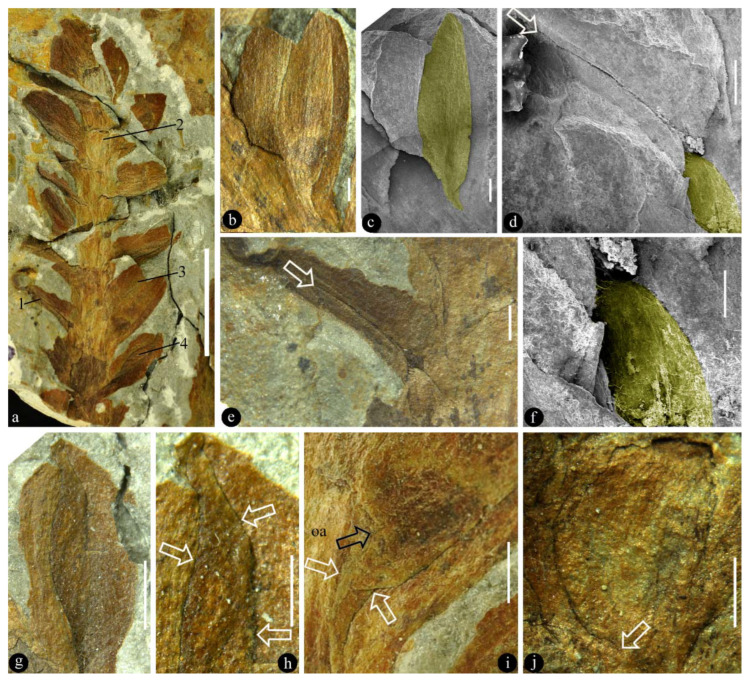
General view and details of the holotype of *Combina* gen. nov. specimen number MPZ2009-425. (**a**) The organ including multiple helically arranged lateral units. Scale bar = 10 mm. (**b**) Detailed view of the lateral unit 3 in (**a**). Scale bar = 1 mm. (**c**) The same as in (**b**), under SEM. The ovule is colored artificially. Scale bar = 1 mm. (**d**) SEM view of the lateral unit 1 in (**a**), showing the adaxial suture (arrow) of the bract. The ovule base is colored artificially. Scale bar = 1 mm. (**e**) View of the lateral unit 1 in (**a**), showing the adaxial suture (arrow) of the bract. Scale bar = 1 mm. (**f**) Detailed view of the basal portion of the ovule (artificially colored) that is eclipsed by the adaxial part of the bract, under SEM, enlarged from (**d**). Scale bar = 0.5 mm. (**g**) View of the lateral unit 4 in (**a**), showing the ovule “sandwiched” adaxially and abaxially by its bract. Scale bar = 2 mm. (**h**) Detailed view of the ovule (arrows), enlarged from (**g**). Scale bar = 1 mm. (**i**) Detailed view of the ovule base, enlarged from (**b**), showing its physical connection (funiculus, between white arrows) to the organ axis (oa). Note the asymmetrical connection (black arrow) between the funiculus and the ovule. Refer to Figure 3a. Scale bar = 1 mm. (**j**) Detailed view of the lateral unit 2 in (**a**), showing the asymmetrical connection (arrow) between the ovule and its funiculus. Refer to Figure 3b. Scale bar = 1 mm.

**Figure 2 plants-11-02833-f002:**
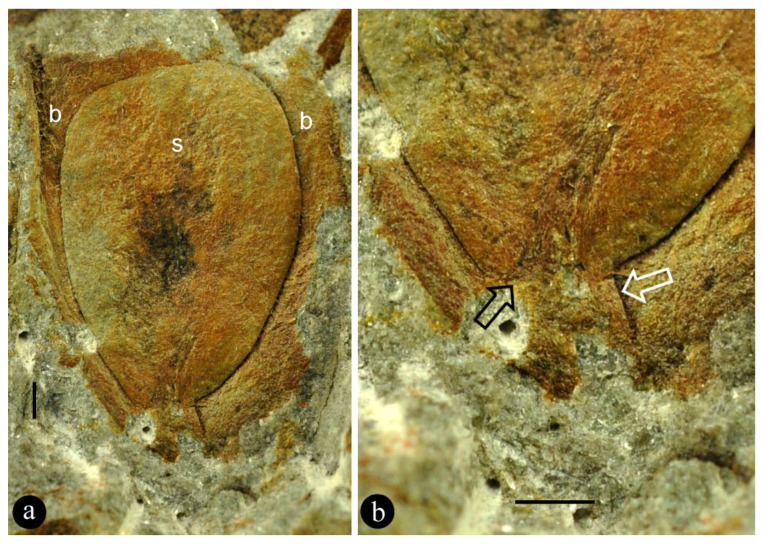
An isolated part associated with the holotype of *Combina* gen. et sp. nov. (**a**) An adaxial view of an anatropous ovule (s) in a bract (b) axil. Specimen number R4/40/38/5. Scale bar = 1 mm. (**b**) Detailed view of the ovule base in (**a**), showing the asymmetrical funiculus (white arrow) and the micropyle (black arrow) of the anatropous ovule. Scale bar = 1 mm.

**Figure 3 plants-11-02833-f003:**
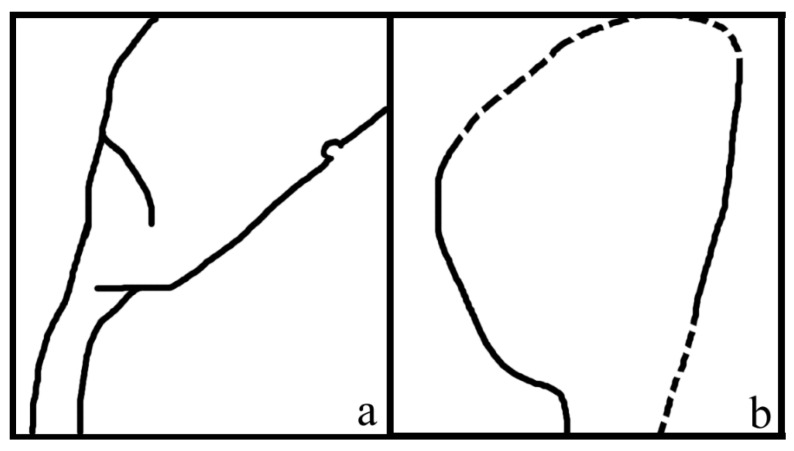
Sketches of *Combina*. (**a**) Asymmetrical ovule base in Figure 1i. (**b**) Asymmetrical ovule base in Figure 1j.

**Figure 4 plants-11-02833-f004:**
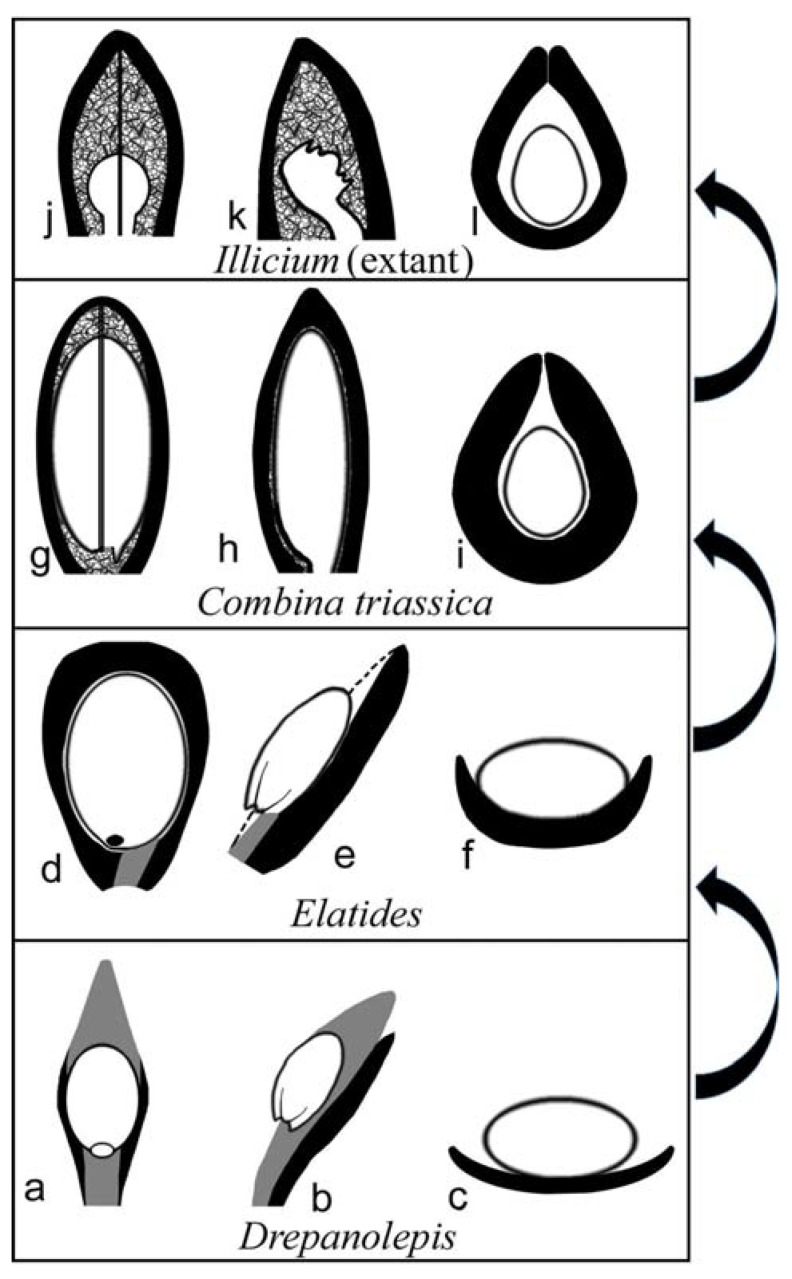
Proposed possible evolutionary roadmap related to *Combina* gen. nov., from a conifer (*Drepanolepis*) to extant *Illicium*. The scale/funiculus is in dark gray, the ovule is in white, and the bract is in black. (**a**–**c**) Adaxial, lateral, and cross view of a lateral unit in *Drepanolepis*. (**d**–**f**) Adaxial, lateral, and cross view of a lateral unit in *Elatides*^23^. Note the wings of the bract curving adaxially. (**g**–**i**) Adaxial, lateral, and cross view of a lateral unit in *Combina* gen. nov. Note the axillary ovule is almost fully enclosed except along the ventral suture (gap). (**j**–**l**) Adaxial, lateral, and cross view of a carpel in extant *Illicium* (Illiciaceae).

## Data Availability

Not applicable.

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
