# Peer review of "Pre-Carpels from the Middle Triassic of Spain"

_plants, 2022, doi:10.3390/plants11212833_

Round 1

Reviewer 1 Report

This is a well written and carefully prepared paper on an interesting fossil from the mid Triassic of Spain. The authors claim that it might be the missing link between conifers and angiosperms and give various arguments for their interpretation. There is however at least one major problem. The transition series given in their fig. 4 ends with Illicium and the authors do not address the evolutionary pathway from uniovultate carpels to multiovulate ones. The inverse pattern is clearly demonstrated in Ranunculaceae and Rosaceae, where the uniovulate ones seem to be derived from the multiovulate ones by reduction to a single one being attached in the transversal zone (“Querzone”). However, both families might show a clear and directed transition series, but are too far from the basis of angiosperms to serve for more than a possible analogy. The unit with two ovules (fig 2c, 3c) is strikingly similar to what is found in many magnolias and these two ovules can easily be interpreted as being attached to the margins of the bract or carpel. This interpretation would be in line with the traditional theory. I have also my difficulties with the term “unifying theory” as this implies or suggests a kind of synthesis between two conflicting concepts, what is clearly not the case. However, the term “unifying theory” has not been coined by the authors and so it is unfair to discuss this in the context of the paper submitted by the authors.

In brief, this are very interesting and important facts. I am not fully convinced about their interpretation, but this has to be discussed in the scientific community and not to be decided by a reviewer. I suggest to accept the paper as it stands.

Please correct p. 4 line 118 “Schweiter” to “Schweitzer”.

Reviewer 2 Report

These are undoubtedly interesting fossils that are worth documenting. However, I am not convinced that they have much to do with angiosperms. Both palaeobotanical and molecular evidence points to the coniferopsid clade separating from the bennettitopsid / gnetopsid clade from in which the angiosperms were probably rooted way back in the Palaeozoic. The idea that we are seeing here evidence of a transition from conifers to angiosperms flies against all other evidence.

As the authors themselves point out, they look very like early coniferalean cones. The Voltziales were, after all, abundant at this time. The main arguments for comparing them with angiosperms is that the bract seems to partly enclose the ovule. But the enclosure of the ovules by the bracts is not really demonstrated in any of the figures, and Fig. 2a in fact seems to show that the ovule was not enclosed. The authors also suggest that there was more than one ovule enlcosed in the bract but again this is not clearly demonstrated; they point to Fig. 2c but this is too poorly preserved to show this convincingly - one of the "ovules" could be merely folding of the bract surface. 

Even if we were to accept that we are seeieng here several ovules being partly enclosed by a bract, this does not make them angiosperms or even ancestral to the angiosperms.

Round 2

Reviewer 2 Report

I have no more substantive comments / suggestions to make at this stage.